# A Systematic Literature Review of Loneliness in Community Dwelling Older Adults

Gabriela Di Perna [1,*], Katrina Radford [2], Gaery Barbery [3] and Janna Anneke Fitzgerald [1]

1   Department of Business Strategy and Innovation, Griffith Business School, Brisbane, QLD 4111, Australia
2   Department of Employment Relations and Human Resources, Griffith Business School,
    Southport, QLD 4215, Australia
3   School of Applied Psychology, Griffith University, 176 Messines Ridge Road, Mt Gravatt, QLD 4122, Australia
*   Correspondence: gabriela.diperna@griffithuni.edu.au

**Abstract:** Research on loneliness is extensive. This paper presents a systematic review of intervention studies, outlining the antecedents to, and consequences of loneliness in community-dwelling older people. Using PRISMA methodology, a systematic literature review was conducted between January and August 2021 resulting in 49 useable articles. Papers were included if they: (a) investigated older people (+50); (b) were living in community dwellings; (c) had been published in English; (d) had titles or abstracts available and, (e) were published between 2016 and 2021. This study found the antecedents and consequences of social, emotional and existential loneliness differ, however, the vast majority of research has not examined the unique types of loneliness and instead kept loneliness as a generic term, despite the acceptance that various types of loneliness exist. In addition, the findings of intervention studies identified through this review have yielded mixed results. Those interventions focused on improving personal and psycho-social resources for older people fared better outcomes than those focused on technological and social connections alone. This paper reports important implications for the future of research conducted on loneliness and interventions accordingly.

**Keywords:** loneliness; older people; community-dwelling older adults; in home care; at home care; CHSP; HCS; home care service*; retire* liv*; indepe*liv*; community dwell*; community care; social*isolat*; lone*; social*exclu*; emotion*isolat*; solitude; se-clu*

## 1. Introduction

The loneliness epidemic exists (Jeste et al. 2020), with 1 in 3 Australians reported experiencing loneliness prior to the lockdowns (AIHW 2021). As lockdowns, isolation and ongoing COVID restraints have increased, the number of people experiencing loneliness and social isolation has increased up to 30% across Europe, USA and China (Galea et al. 2020; Hwang et al. 2020; Jeste et al. 2020; McGinty et al. 2020).

Loneliness has been associated with higher risks of mortality (Holt-Lunstad et al. 2015), coronary heart disease, and stroke (Valtorta et al. 2016). Importantly, loneliness has been identified as "geriatric giant" (Routasalo and Pitkala 2003, p. 303) negatively impacting the physical and mental health of older people, affecting their quality of life. While seniors are living longer than ever before, the quality of life in later years is declining, particularly in terms of mental health (AIHW 2021). Older persons are particularly at risk of dementia, cognitive decline (Zhong et al. 2018), and depression (Cacioppo et al. 2002).

Hence, research studies that aim to reduce loneliness in older adults are in demand.

Fortunately, much research has already been published recently on the risk and protective factors of loneliness (Philip et al. 2020; Teater et al. 2021). However, no study has combined them to explore not only the antecedents and consequences of loneliness, but also reviewed the effectiveness of interventions that aimed to reduce loneliness in older adults living in the community. This study was designed to address these gaps, focusing on the antecedents and consequences of loneliness.

Loneliness is generally considered to be subjective in nature, and includes negative feelings related to being alone and reduced social connections (Weiss 1973). However, it can also be conceptualised as multilayered with three types of loneliness emerging from the literature: social loneliness, emotional loneliness, and existential loneliness (Weiss 1975). Social loneliness results from limited social networking and engaging with contacts and can be quantifiably measured through a reduced social network (Valtorta and Hanratty 2012). Emotional loneliness is the feeling of lacking close and intimate relationships. While existential loneliness involves "the immediate awareness of being fundamentally separated from other people and from the universe, primarily through experiencing oneself as mortal, or, and especially when in a crisis, experiencing not being met at a deep human (i.e., authentic) level" (Bolmsjö et al. 2019, p. 5). That is, existential loneliness is deeply rooted in the feeling of existence and is affected by feelings of both social and emotional loneliness.

Research to date has largely explored the antecedents and consequences of loneliness separately, as well as, exploring interventions that reduce older adults' feelings of loneliness. This study extends the current knowledge by conducting a systematic literature review to explore loneliness in older adults as an overarching concept, including studies looking at antecedents and consequences as well as the effectiveness of loneliness interventions on community dwelling older adults. This study explored two research questions:

(1)　What are the antecedents and consequences of loneliness?
(2)　What are the effects of interventions aimed at reducing loneliness within community dwelling older adults?

## 2. Methods

### 2.1. Aim

To identify and synthesise studies on loneliness in older community-dwelling adults with the aim to identify the available evidence reporting on the antecedents and consequences of loneliness as well as intervention studies that targeted reducing loneliness in community-dwelling older adults.

### 2.2. Design

A systematic review was performed between January and August 2021 and captured empirical studies completed in the past 5 years to capture contemporary evidence on loneliness in older community-dwelling adults. For rigor, the study used the Preferred Reporting Items for Systematic Reviews and Meta-Analyses framework (Page et al. 2021).

A search strategy was developed and modified for each of the following databases: MEDLINE, SCOPUS, CINAHL and PsychInfo. The search strategy focused on, firstly the community-dwelling environment of the older adult (home care, home care services, commonwealth home support program (CHSP) and retirement living/independent living), and secondly, the experiences of loneliness of the elder (social isolation, loneliness, social exclusion, emotional isolation, solitude, and seclusion). Loneliness was searched as a general term to include all existing types of loneliness that might have been identified within the literature. The third search combined the two previous searches. MeSH search terms were informed by an experienced university librarian skilled in systematic reviews. Social isolation was included in the search term as until more recently, social isolation has been used interchangeably with loneliness. Thus, to be inclusive of all possible citations referring to loneliness, we added social isolation to our search list. While a formal protocol paper was not prepared in advance of this paper, the search terms are detailed as follows:

1.　"in home care"OR "at home care"OR "CHSP"OR "HCS"OR "home care service*OR "retire* liv*" OR "indepe*liv*" OR "community dwell*" OR "community care"
2.　"social*isolat* OR lone* OR social*exclu* OR emotion*isolat*OR solitude OR seclu*"
3.　1. AND 2.

Empirical studies were included if they: (a) investigated older adults (+50); (b) were living in community dwellings; (c) had been published in English; (d) had titles or ab-

stracts available (e) were published between 2016 and 2021 and (f) explored loneliness as a construct of interest.

A total of 876 citations were identified through the databases searched. Once duplicates were excluded (n = 409), the remaining publications titles and abstracts (n = 467) were screened by two researchers (GDP and GB) individually and then agreeing on the findings, to exclude 369 studies based on the selection criteria. The remaining 98 were further screened by reading the full texts by the same two researchers (GDP, GB). In cases of any differences in evaluations, a third researcher (AF) independently evaluated these studies, and a consensus was reached among the researchers. At the end of the selection process, 49 publications were included (see Figure 1).

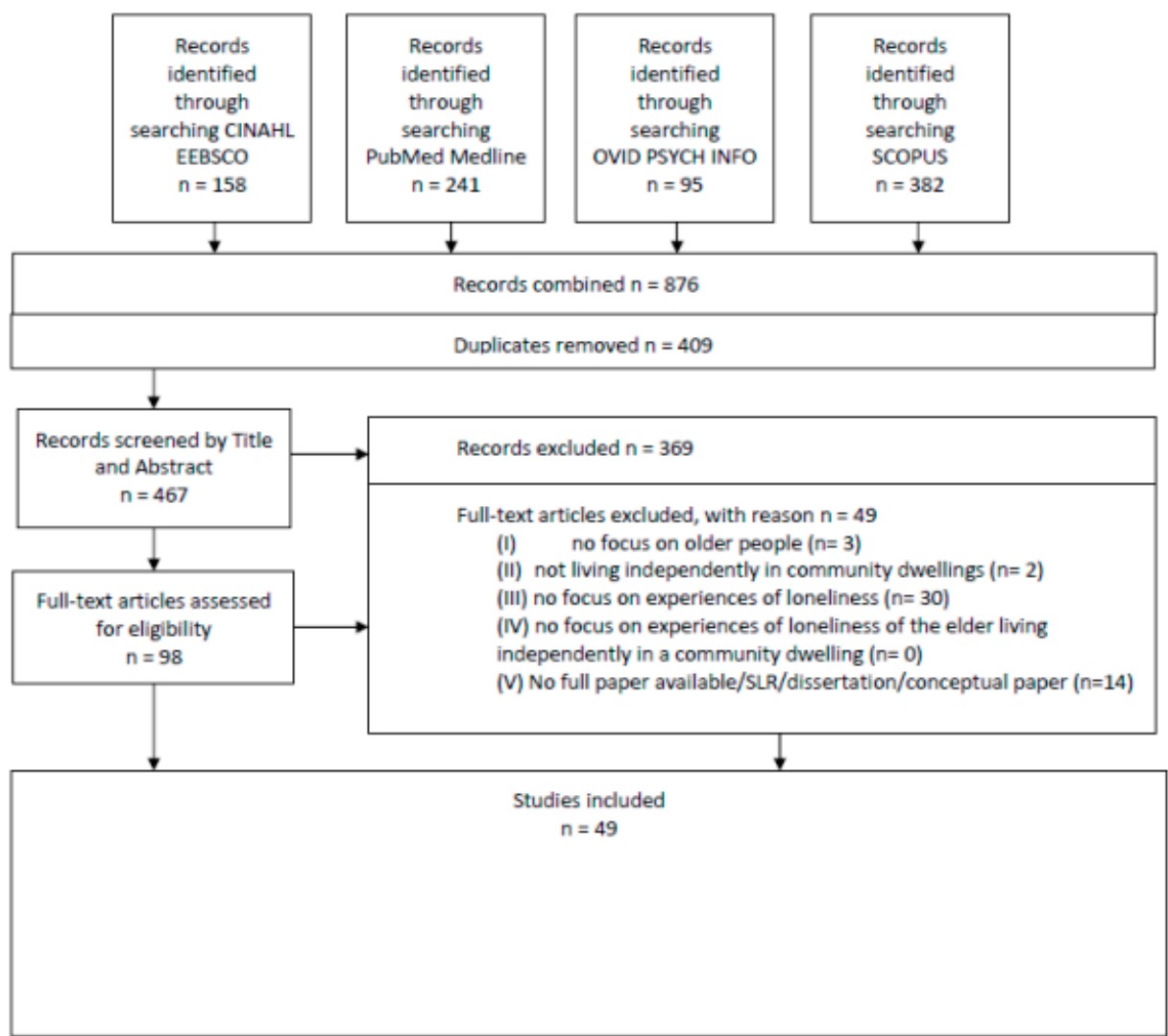

**Figure 1.** PRISMA flow diagram of information through different phases of the systematic review (Liberati et al. 2009).

### 2.3. Quality Appraisal

The included studies were evaluated for their methodological quality via the Mixed-Methods Appraisal Tool (MMAT, Hong et al. 2018). The MMAT tool can be used to assess both mixed-method studies as well as, qualitative and quantitative studies. MMAT does

not differentiate between qualitative, quantitative, and mixed method studies in terms of its ranking. The checklist has 21 criteria, divided over six categories: (a) screening questions (for all types); (b) qualitative; (c) quantitative randomised controlled trials; (d) quantitative nonrandomised; (e) quantitative descriptive; and (f) mixed methods. For every item, there can be three responses: 'yes', 'no', or 'can't tell' (Hong et al. 2018). The number of items rated "yes" were counted to provide an overall score of each publication. Percentages of MMAT outcomes were calculated to compare the methodological quality of the included articles. Two researchers (GDP, GB) assessed the quality of the selected publications independently and any discrepancies were resolved through a third researcher (AF). The quality of the included studies was assessed to explore their possible contribution to the synthesis. Thus, while methodological quality was not considered as an exclusion criterion in this study, it was used as a guide when interpreting and weighing up the findings of the studies included (Bettany-Saltikov 2012).

### 2.4. Data Synthesis

Content analysis was used to synthesis and analyse the data. The content of each article was read and coded to highlight concepts that were raised from the study. Then, these codes were constantly compared with the findings of the other selected studies for the purpose of identifying common themes and conceptual categories. At the end of this analytical stage, the categories emerged from the studies were grouped according to their similarities into overarching themes, as shown in the results section.

## 3. Results

A total of 49 articles were retained for analysis. While some antecedents for social and emotional loneliness overlap in part, the authors found it important to report on each type separately in order to create a better understanding of the patterns that emerge.

### 3.1. Antecedents to Loneliness

From the total of 49 articles examined, 35 studies actively discussed the antecedents to loneliness, which were broken up into emotional, social and existential loneliness, as well as general loneliness (where no specific aspects of loneliness were identified in the paper). Of these, 4 papers measured social loneliness, 4 papers measured emotional loneliness, and 2 papers measured existential loneliness. The remainder (n = 25) treated loneliness as a general concept. Consequently, a review of each is presented below. Figures 2–4 highlight the active relationship pathways identified by these papers as directly and indirectly impacting loneliness.

### 3.2. Emotional Loneliness

Figure 2 depicts the antecedents to emotional loneliness. As highlighted, older age (Fierloos et al. 2021; Gibney et al. 2019), females who were living with a partner (Evans et al. 2019; Fierloos et al. 2021; Gibney et al. 2019; (Nieboer et al. 2020) those who had lower educational levels (Fierloos et al. 2021; Gibney et al. 2019) and cultural background (Nieboer et al. 2020) were identified as reporting higher levels of emotional loneliness. Interestingly these results differed across cultures. For example, in Croatia living without a partner was not associated with increased emotional loneliness but in the UK, Greece, The Netherlands and Spain, it was (Fierloos et al. 2021). Participants from Greece, Croatia and The Netherlands had a significantly higher odds ratio of reporting emotional loneliness than those from Spain, with participants from Greece reporting the highest odds ratio. Activities that involved the self-management of loneliness were crucial in protecting people from experiencing emotional loneliness. For example, participants who took initiative to meet new people, invested in their own personal behaviour around social connections, had a variety of resources available to them, and could multifunction the use of resources they had available to them showed lower levels of emotional loneliness scores according to a study by Nieboer et al. (2020).

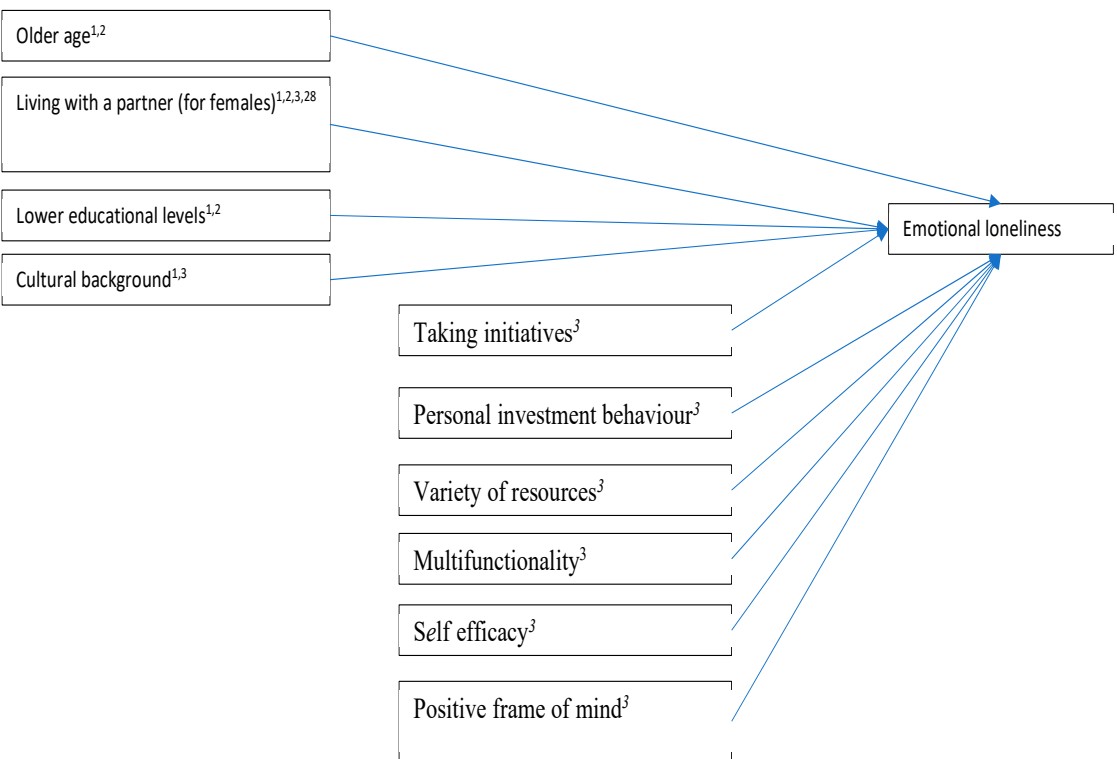

**Figure 2.** Antecedents to emotional loneliness. 1. Fierloos et al. (2021); 2. Gibney et al. (2019); 3. Nieboer et al. (2020); 28. Evans et al. (2019).

### 3.3. Social Loneliness

Figure 3 shows the antecedents for social loneliness (Fierloos et al. 2021; Gibney et al. 2019; Nieboer et al. 2020). Males living without a partner, reported a 1.4 times higher odds ratio of experiencing social loneliness than females (Fierloos et al. 2021), however in general advanced age (Gibney et al. 2019; Nieboer et al. 2020), those participants with a lower educational background (Fierloos et al. 2021; Nieboer et al. 2020), who were unemployed (Gibney et al. 2019) and reported poor health (Gibney et al. 2019; Nieboer et al. 2020), were also more likely to experience social loneliness. Like emotional loneliness, differences were found across cultures, for example those living in Greece, Croatia and the Netherlands had a significantly higher odds to experiencing social loneliness (Fierloos et al. 2021). Participants living in Croatia were 8.34 times more likely to report experiencing social loneliness, compared to participants from Greece (1.88 times), and the Netherlands (1.66 times). However, the interaction between culture and living situation was not significant for social loneliness as it was for emotional loneliness (Fierloos et al. 2021). Multifunctionality, having a positive frame of mind, taking initiative and self-efficacy were also noted as antecedents to social loneliness (Evans et al. 2019).

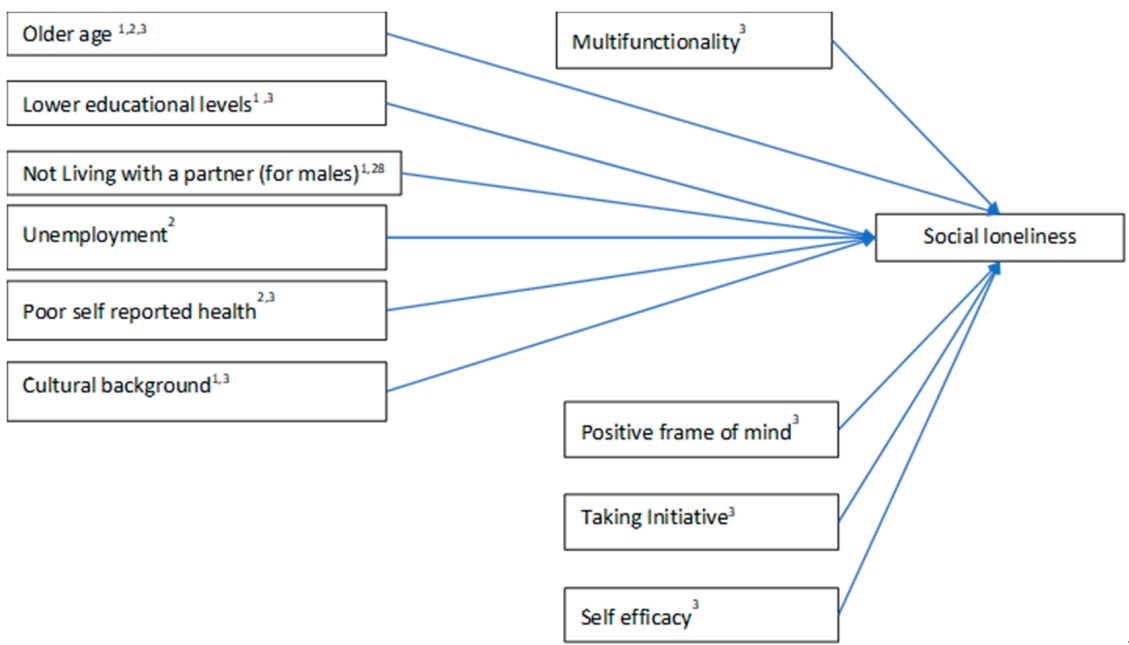

**Figure 3.** Antecedents to social loneliness. 1. Fierloos et al. (2021); 2. Gibney et al. (2019); 3. Nieboer et al. (2020); 28. Evans et al. (2019).

### 3.4. Existential Loneliness

The antecedents to existential loneliness were measured by only two studies, as evident in Figure 4 (Sjöberg et al. 2019; Hemberg et al. 2019). Poor health was found to predict existential loneliness in one of these (Hemberg et al. 2019) and existential loneliness was found to be eased by being acknowledged by others, experiencing meaningful togetherness with others and themselves, bracketing negative thoughts, connecting to partners, or loved ones and, continuing social relationships (Sjöberg et al. 2019; Hemberg et al. 2019).

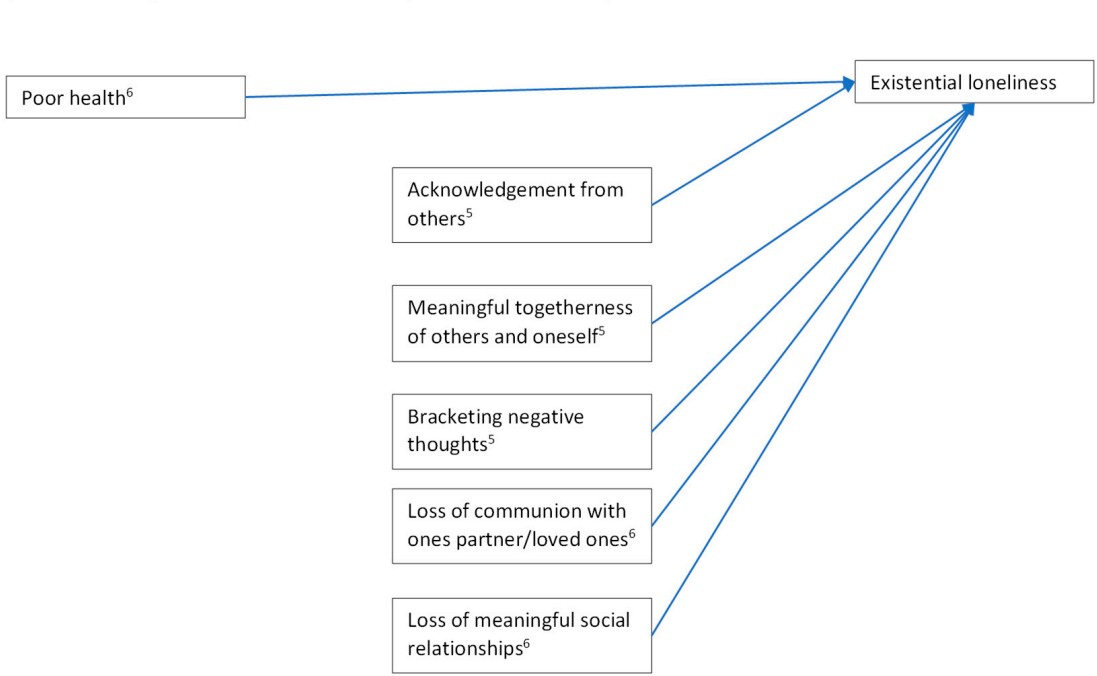

**Figure 4.** Antecedents to existential loneliness. 5. Sjöberg et al. (2019); 6. Hemberg et al. (2019).

### 3.5. General Loneliness

Most articles studied overall loneliness in participants. In these studies, most authors used either a general question, open ended interview questions, or the UCLA (and its cultural adaptation measures) 3-item measure of loneliness across the globe, and across rural, remote, regional, and metropolitan locations. Resulting in the factors listed in Figure 5 being presented.

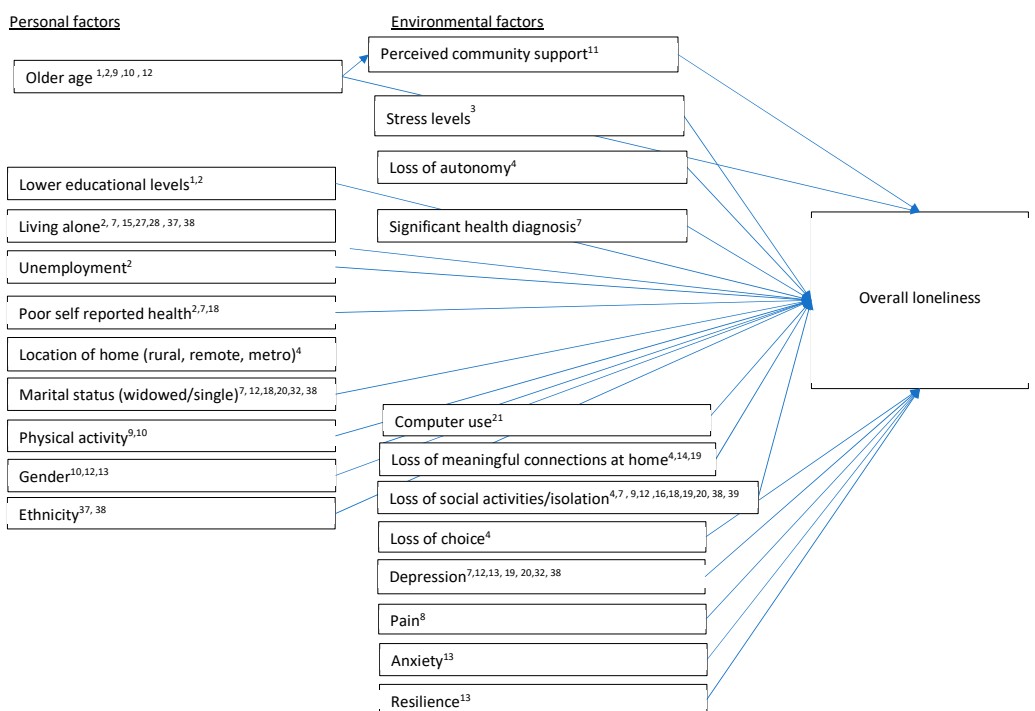

**Figure 5.** Antecedents to general loneliness. 1. Fierloos et al. (2021); 2. Gibney et al. (2019); 3. Nieboer et al. (2020), 4. Herron et al. (2022); 7. Fernandes et al. (2018); 8. Emerson et al. (2018); 9. Lin et al. (2016); 10. Gyasi et al. (2021); 11. Park et al. (2019); 12. Tomstad et al. (2017); 13. Lee et al. (2019); 14. Bai et al. (2017); 15. Teguo et al. (2016); 16. Wong et al. (2017); 17. Lay et al. (2020); 18. Teater et al. (2021); 19. Warner et al. (2019); 20. Ojagbemi et al. (2021); 21. Talmage et al. (2021) 27. Gyasi et al. (2019), 28. Evans et al. (2019); 32. Lam et al. (2017); 37. Jamieson et al. (2019); 38. Cheung et al. (2019).

In general, studies reported that advancing age, lower educational levels, living alone, being unemployed, poor self-rated health, being widowed/single, having limited physical activity and being female were all associated with reporting higher levels of loneliness within the community. Moreover, the relationship between age and overall loneliness was mediated by perceived community support in Chinese older people in a study by Park et al. (2019). However, while cultural background appeared to influence the level of emotional and social loneliness experienced in general studies of loneliness, culture was not found to influence loneliness (Lin et al. 2016).

Factors, such as, having a loss of autonomy, meaningful connections, social activities/connections, lower technology usage and choice were all social factors influencing loneliness scores across multiple studies. A loss of social connections was reported as the most influential in those who reported loneliness. A study by Wong et al. (2017) also found structural drivers of social alienation led to loneliness in their qualitative study of older people in Hong Kong. In addition, a study of community dwelling older people in Canada and Hong Kong, found older people who existed in isolation to others or who did not engage in social activities were at a higher risk of loneliness (Lay et al. 2020).

Mental health conditions, such as, high stress levels, anxiety, and lower levels of resilience were associated with higher reported perceived loneliness. Experiencing pain and being diagnosed with a significant health condition (identified as poor self-reported

health in Figure 5), such as cancer, was also reported to influence feelings of loneliness by participants in these studies. With the odds of reporting loneliness being 1.58 times higher for those experiencing long term pain through reporting pain levels in both the 2008 and 2012 surveys of the health and retirement study data in the United States of America (Emerson et al. 2018).

These findings are significant as they suggest that loneliness, whether general, social, emotional, or existential have different antecedents and as such, interventions should be targeted accordingly. In addition, it provides insights on "who" is at risk of loneliness within the community and as such allows researchers to target interventions appropriately.

### 3.6. Consequences of Loneliness

A total of 14 articles focused on the consequences of loneliness. The findings of these studies highlighted that loneliness has a significant impact on physical and mental states of older adults living in the community, as highlighted in Figure 6.

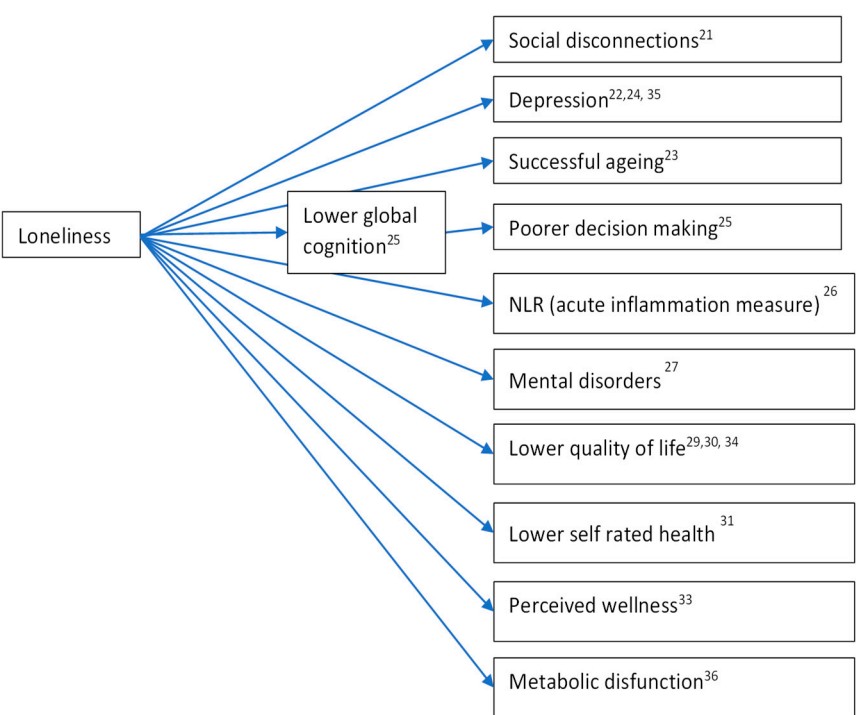

**Figure 6.** Consequences of loneliness. 21. Talmage et al. (2021); 22. Gonyea et al. (2018); 23. Salamene et al. (2021); 24. Warner et al. (2019); 25. Stewart et al. (2020); 26. Koyama et al. (2021); 27. Gyasi et al. (2019); 29. Ward et al. (2019); 30. Schorr and Khalaila (2018); 31. McHugh and Lawlor (2016); 33. Hodgkin et al. (2018); 34. Tan et al. (2020); 35. McHugh Power et al. (2020); 36. Shiovitz-Ezra and Parag (2019).

As evident in Figure 6, loneliness was found to result in higher levels of social disconnection (Talmage et al. 2021), depression (Gonyea et al. 2018; McHugh Power et al. 2020; Warner et al. 2019) mental disorders (Gyasi et al. 2019), lower quality of life (Schorr and Khalaila 2018; Tan et al. 2020; Ward et al. 2019) lower self-rated health scores (McHugh and Lawlor 2016), perceived wellness (Hodgkin et al. 2018), as well as, metabolic disfunction (Shiovitz-Ezra and Parag 2019) and acute inflammation (Koyama et al. 2021). Whereas the relationship between loneliness and decision making was mediated by lower global cognition (Stewart et al. 2020). The presence of poor self-rated health, mental disorders, and depression outcomes in research exploring both the antecedents and consequences of loneliness suggest these relationships are complex but interconnected.

What was also interesting in these results was, while most studies explored loneliness as a general concept, the two studies who separated loneliness into the different types of

loneliness found differences within the relationships. For example, Tan et al. (2020) explored general loneliness, emotional and social loneliness factors and found that while loneliness was associated with lower mental Health Related-Quality of Life (HR-QoL), emotional loneliness had a stronger association with mental HR-QoL than social loneliness or physical HR-QOL. Similarly, McHugh Power et al. (2020) found a stronger pathway between emotional loneliness and depressive symptoms than social loneliness and depressive symptoms. These findings provide additional support for the need to measure different types of loneliness going forward.

After reviewing the evidence for the antecedents and consequences of loneliness, this study then moved onto exploring the evidence of effectiveness of interventions targeting loneliness.

### 3.7. Outcomes of Interventions for Loneliness

A total of 10 studies were identified that focused on reporting outcomes of interventions focused on addressing loneliness in community dwelling older people. Of these interventions, only one study differentiated between types of loneliness, whereas the remaining nine articles focused on reducing general feelings of loneliness.

Of the ten intervention studies identified, only four reported significant changes to loneliness as an outcome variable. Those interventions that did see a reduction in loneliness seemed to concentrate on producing strong psycho-social connections (such as relationship with peers) and strategies to participants throughout their interventions. In addition, while it was difficult to evaluate the outcomes of these programs due to a lack of detail about the program that was run, five of the six successful programs, defined as those influencing loneliness positively, focused on developing meaningful relationships between peers, or with peers, during the program (Geffen et al. 2019; Lai et al. 2020; Hwang et al. 2019; Ristolainen et al. 2020; Rodríguez-Romero et al. 2021; Shapira et al. 2021). The sixth program focused on gratitude activities that were found to not only decrease loneliness but also boost self-reported health outcomes (Bartlett and Arpin 2019).

Of the remainder of interventions, while they were not successful in reducing loneliness, they did provide unique insights into intervention programs for the older community-dwelling people involved. For example, Kharicha et al. (2017) conducted a study to explore the perceptions and experiences of older people on community-based avenues of support programs in England and found that to attract older participants in the community, there is a need to ensure the program connects those with a common interest and that the program does not have any stigma attached to it. That is, participants in their study reported that 'befriending' programs carried a social stigma of loneliness, and they did not want to be seen as lonely in front of their peers. However, programs that focused on connecting individuals with likeminded interests were attractive.

Moreover, the studies conducted by both Gustafsson et al. (2017) and Rolandi et al. (2020) produced interesting findings as, while no differences in loneliness were identified post the intervention programs, both programs were run for 4 or 5 weeks whereas the more successful programs had a longer duration, indicating that the intervention timing itself may be important for outcomes.

## 4. Discussion

Research on loneliness is extensive, and there is an acceptance that loneliness can be distinguished between emotional, social, and existential loneliness (Bolmsjö et al. 2019; Valtorta and Hanratty 2012; Weiss 1973). This paper was designed to systematically review loneliness in community dwelling older adults in general. In doing so, most programs included in this review (35) focused on antecedents of loneliness, while only 14 looked at the consequences of loneliness. Furthermore, while every effort has been made to unpack the effectiveness of interventions focused on loneliness, this review was based on only 10 articles, that discussed loneliness intervention in community dwelling older people. In doing so, this study found that despite the acceptance of the different types of loneliness

and the availability of tools to measure these types, most research on loneliness only focuses on loneliness as a generic term. Yet, this study has also highlighted that the antecedents for the different types of loneliness differ, as do the consequences.

Age, living situation, lower education levels, cultural factors, having a positive frame of mind, multifunctionality, taking initiative and self-efficacy were similar antecedent factors predicting both social and emotional loneliness. For those who reported experiencing social loneliness, unemployment and poor self-reported health were reported as additional antecedents (Fierloos et al. 2021; Gibney et al. 2019; Nieboer et al. 2020; Evans et al. 2019). In addition, those who experienced social loneliness reported having no multifunctionality (the ability to multi-task), have a negative frame of mind, did not take initiative with social networks, and had a lower self-efficacy score (Fierloos et al. 2021; Gibney et al. 2019; Nieboer et al. 2020; Evans et al. 2019). Interestingly when it came to the consequences of these types of loneliness, this study highlighted that emotional loneliness had a stronger correlation to Health Related-Quality of Life (Tan et al. (2020), and a stronger correlation to depressive symptoms than social loneliness (McHugh Power et al. 2020).

These findings are important as it suggests that interventions aimed at influencing social and emotional loneliness outcomes need to be targeted accordingly. Yet, the interventions identified in this study to reduce loneliness were found to only target general loneliness and had mixed outcomes. Consequently, future research is needed to focus interventions that improve older peoples' psycho-social behavioural resources to influence social and emotional loneliness outcomes.

This study also found that two studies examined the antecedents to existential loneliness and found that poor health, loss of meaningful social relationships, loss of communion with partners or loved ones, the lack of bracketing negative thoughts, a lack of acknowledgement from others, and a lack of meaningful togetherness of others and oneself influences older peoples' perception of existential loneliness.

The finding that the antecedents and consequences for social, emotional and existential loneliness are different is not necessarily surprising given the definition and scope of the types of loneliness (Valtorta and Hanratty 2012). What is surprising is the lack of research that individually explores these concepts separate to the construct of loneliness as an overarching general construct, particularly given the acceptance of the terms within the literature for the past 20 years (Van Tilburg 2021, 2022) and the limitation of this study to research within the last 5 years. Thus, this paper is perhaps an important step in the literature to return to the importance of measuring the different types of loneliness rather than loneliness in general.

This paper importantly elucidates the complex two-directional relationship of psycho-social outcomes of stress (Nieboer et al. 2020), depression (Fernandes et al. 2018; Tomstad et al. 2017; Ojagbemi et al. 2021), anxiety (Lee et al. 2019) and social isolation (Herron et al. 2022; Talmage et al. 2021) with loneliness. Where these variables are found as important antecedents to loneliness in some cases, and in other studies they are identified as consequences of loneliness (Talmage et al. 2021; Gonyea et al. 2018; Warner et al. 2019). In addition, this study highlighted that loneliness not only impacts psycho-social variables such as depression, successful ageing, social disconnection, decision making, quality of life and perceived health, but also impacts physical health outcomes of metabolic disfunction, perceived wellness, and acute inflammation measures (Koyama et al. 2021). These findings emphasise the importance of addressing loneliness in older adults living in the community through exploring interventions that may assist in reducing the different types of loneliness.

When examining the effectiveness of interventions aimed at reducing loneliness, the findings were mixed, however those studies that equipped participants with psycho-social skills to address their loneliness tended to be more successful than those who did not. Simply interacting with others was found to not be enough to make a difference, instead interventions are required that connect like-minded people or a shared interest group if they are to be effective.

An interesting observation from this study, is that methodologically, the majority of studies included in the review of interventions involved general populations of interest, rather than targeting those who were lonely, with the exception of one paper that targeted identified people through their General Practitioner. Given the importance of continuing the research on loneliness in older people, it is possible that the factors identified as antecedents in this paper could serve as a potential guide for the inclusion of older people in the community in future studies.

As the COVID-19 pandemic continues, and society goes in and out of lockdowns, it is important to equip older people living in the community with psycho-social skills to connect to others, as well as, with those skills designed to improve their resilience, self-efficacy and personal resources. This paper provided an important pause in the literature for reflection about how we design interventions for older adults living in the community and also, how we better research the different types of loneliness to combat the loneliness epidemic.

While this paper only conducted a systematic review of the loneliness literature in the past 5 years, in doing so it highlighted important areas for future research to focus on. Without more focused research on the different types of loneliness it is unlikely that we are going to address the growing loneliness epidemic within our society.

**Author Contributions:** Conceptualization, J.A.F., G.D.P., K.R. and G.B.; methodology J.A.F., G.D.P., K.R. and G.B.; software, G.B. and J.A.F.; validation, G.D.P.; formal analysis, K.R.; investigation, J.A.F., G.D.P. and G.B.; resources, J.A.F., G.D.P., K.R. and G.B.; data curation, J.A.F., G.D.P., K.R. and G.B.; writing—original draft preparation, K.R.; writing—review and editing, J.A.F., G.D.P., K.R. and G.B.; visualization, K.R.; supervision, J.A.F., G.B. and K.R.; project administration, G.D.P.; funding acquisition, J.A.F., G.D.P., K.R. and G.B. All authors have read and agreed to the published version of the manuscript.

**Funding:** This project was funded by the Cromwell Property Group Foundation: 212387 1301 00000.

**Institutional Review Board Statement:** This project was underpinned by Griffith University Human Resource Ethics 2022/166.

**Informed Consent Statement:** Not applicable.

**Conflicts of Interest:** The authors declare no conflict of interest.

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
