# Peer review of "A Systematic Literature Review of Loneliness in Community Dwelling Older Adults"

_socsci, doi:10.3390/socsci12010021_

Round 1

Reviewer 1 Report

Please, check the attachment.

Reviewer 2 Report

Considering the high levels of loneliness verified and the verification of this problem as a public health problem, I believe that the subject of this manuscript is interesting, and the results presented are clearly and concisely. However, in the discussion section authors may include other bibliographic references included in the review. Despite being clear and interesting, the discussion seems to be a repetition of the results, as there is a lack of dialogue with other works on the revealed findings.

Round 2

Reviewer 1 Report

The article was adapted according to the reccomendations therefore I suggest to publish the article in newly form.

Kind regards